# Learning Interpretable Representations Leads to Semantically Faithful EEG-to-Text Generation

## Abstract

Pretrained generative models have opened new frontiers in brain decoding by enabling the synthesis of realistic texts and images from non-invasive brain recordings. However, the reliability of such outputs remains questionable—whether they truly reflect semantic activation in the brain, or are merely hallucinated by the powerful generative models. In this paper, we focus on EEG-to-text decoding and address its hallucination issue through the lens of posterior collapse. Acknowledging the underlying mismatch in information capacity between EEG and text, we reframe the decoding task as semantic summarization of core meanings rather than previously verbatim reconstruction of stimulus texts. To this end, we propose the Generative Language Inspection Model (GLIM), which emphasizes learning informative and interpretable EEG representations to improve semantic grounding under heterogeneous and small-scale data conditions. Experiments on the public ZuCo dataset demonstrate that GLIM consistently generates fluent, EEG-grounded sentences without teacher forcing. More importantly, it supports more robust evaluation beyond text similarity, through EEG-text retrieval and zero-shot semantic classification across sentiment categories, relation types, and corpus topics. Together, our architecture and evaluation protocols lay the foundation for reliable and scalable benchmarking in generative brain decoding.

## 1 Introduction

Brain decoding lies at the intersection of neuroscience and engineering, offering a path to understanding how the brain encodes perceptual and cognitive states, as well as the foundation for building brain-computer interfaces (BCIs) (Haynes & Rees, 2006; Mathis et al., 2024). Traditionally, decoding has relied on discriminative models that predict labels or stimulus properties from simultaneously recorded brain functional activity (Yamins et al., 2014; Défossez et al., 2023). While effective for constrained tasks, such approaches are inherently confined by closed label sets and offer limited insight into the richness of internal representations (Luo et al., 2023; Benchetrit et al., 2024). With recent success of large-scale generative models in multimodal learning, brain decoding is undergoing a paradigm shift—from discriminative to generative brain decoding, where the goal is to generate structured, naturalistic outputs (e.g., images and texts) directly from brain signals (Chen et al., 2023; Takagi & Nishimoto, 2023; Tang et al., 2023; Wang & Ji, 2022). This generative paradigm facilitates open-ended exploration of neural semantics and enables flexible, expressive brain-computer communication beyond classification or retrieval (Benchetrit et al., 2024).

A promising instantiation of generative brain decoding is the EEG-to-text task, which pairs two modalities with desirable properties: Electroencephalogram (EEG) offers a non-invasive, low-cost input suitable for large-scale data collection (Edelman et al., 2024), while text (i.e., language) provides a semantically rich and compositional output space—serving as the default medium for interpreting meaning in both human mind (Mahowald et al., 2024) and multimodal models (Han et al., 2024). Recent studies have explored this task using sequence-to-sequence models that translate EEG signals to full sentences, typically by conditioning pretrained language models on EEG inputs recorded during natural reading tasks (Murad & Rahimi, 2024). However, these approaches predominantly rely on teacher forcing and evaluate outputs using surface-level text similarity metrics (Wang & Ji, 2022; Duan et al., 2023; Wang et al., 2024), which may not reliably indicate

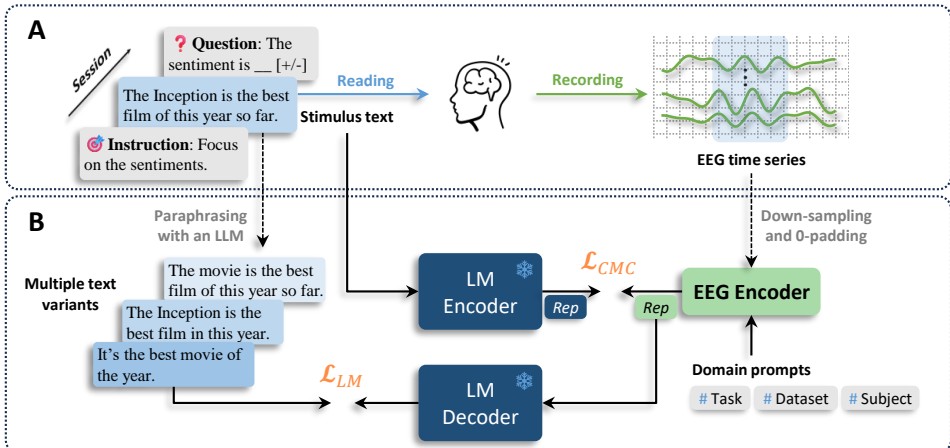

Figure 1: **Overview of GLIM. (A)** One typical experimental session in natural reading dataset (Hollenstein et al., 2018) involves a task-specific instruction followed by sentence stimulus blocks and comprehension queries. Participants read at their own speed and the simultaneously recorded EEG signals are segmented to aligned with each sentence, forming the EEG-text pairs for downstream decoding studies. We consider that several factors—including task-specific goals, uneven attention, and the limited signal-noise ratio (SNR) of EEG—can collectively introduce non-negligible information mismatch between stimulus texts and EEG signals; while the small-scale data and domain heterogeneity further challenge data-driven models to converge and generalize. **(B)** GLIM acknowledges these practical constrains and reframes EEG-to-text as summarization task, targeting semantically faithful rather than syntactically mimetic sentence generation. It focus on the effectiveness and interpretability of EEG decoding in heterogeneous dataset, approached by end-to-end learning informative EEG representations that well-aligned with the high-level representations of a fully frozen pretrained language model.

semantic grounding in the EEG input. Notably, recent analyses have revealed that these models can produce plausible outputs even from random-noise inputs, suggesting weak alignment between EEG inputs and the generated texts (Jo et al., 2024). We argue that this phenomenon reflects a broader challenge known as posterior collapse in generative modeling (Van Den Oord et al., 2017), where the powerful language decoders practically hallucinate full sentences from their language priors and the generic textual pattern of all stimulus texts—instead of conditioning on the semantic information decoded from EEG inputs to perform semantically faithful generation (see Sec. 2 for more details about the posterior collapse phenomenon).

To address the posterior collapse in EEG-to-text decoding, we propose the Generative Language Inspection Model (GLIM)—a framework that reframes the decoding task as semantic summarization. Rather than pursuing word-by-word stimulus reconstruction, GLIM aims to extract and express the core meaning of sentences encoded in EEG signals. This reframing consider the abstract, lossy, heterogeneous nature of mental representations captured by EEG signals, and acknowledges the scale, distribution limitations in current datasets (see Fig. 1). Specifically, GLIM integrates three targeted innovations: (1) a contrastive-generative objective that aligns EEG representations with a frozen language model's latent space, regularizing autoregressive language modeling and providing robust semantic supervision; (2) multiple paraphrased variants of each stimulus text, which augment training data to promote semantic robustness and reduce overfitting; and (3) domain-prompt injection along with minimized, unified EEG preprocessing strategy, enabling robust joint training across heterogeneous domains. Together, these components enable GLIM to effectively decode core semantics from heterogeneous, noisy EEG data while supporting direct inspection on both the generated texts and intermediate representations.

We evaluate GLIM on the ZuCo dataset (Hollenstein et al., 2018; 2019), which provides EEG recordings collected during natural English sentence reading tasks with associated sentence-level annotations. GLIM demonstrates strong semantic decoding performance across cross-modal retrieval and zero-shot classification, and reliably generates coherent sentences grounded in EEG input. Our contributions are threefold:

- **Task reframing**: Based on our identification of posterior collapse as the core failure mode, we reinterpret EEG-to-text decoding as a summarization task, aligning the decoding objective with the abstract and noisy nature of EEG signals; and the dataset limitations in scale and corpus diversity.

- **Scalable architecture**: We present a modular, plug-and-play modeling framework with minimal preprocessing and parameter overhead, enabling the adaptive learning of informative EEG representations and supporting seamless scaling of EEG-to-text decoding across both model capacity and data domains.

- **Zero-shot semantic evaluation**: We establish quantitative evaluation protocols for EEG representations, including EEG-text retrieval and zero-shot classification of high-level semantic categories—without requiring semantic labels during training—supporting interpretable analysis and open-vocabulary semantic decoding.

## 2 RELATED WORK

**Hallucination and posterior collapse.** Hallucination is a foundational challenge across generative models, referring to the generation of fluent and plausible content that fails to follow input instructions or reflect the factual information (Rawte et al., 2023; Ji et al., 2023). It is particularly evident in modern multimodal models (Li et al., 2023b; Bai et al., 2024; Jesson et al., 2024), and increasingly recognized as the primary obstacle in generative brain decoding (Jo et al., 2024; Mayo et al., 2024; Shirakawa et al., 2024). In EEG-to-text decoding, Jo et al. (2024) reproduced the *EEG2Text* model (Wang & Ji, 2022) and observed two symptoms: (1) plausible sentence generation from random noise inputs, and (2) repetitive default outputs (e.g., "He was...") when teacher forcing was disabled. We recognize that these symptoms are closely related to posterior collapse (Van Den Oord et al., 2017; Goyal et al., 2017), where the noisy inputs are ignored as powerful autoregressive decoders directly model the outputs, which leads to above failure in extracting meaningful information from EEG signals. While originally studied in variational autoencoders (VAEs), posterior collapse can broadly account for hallucination in current multimodal models—many of which share structural traits such as encoder-decoder architecture, information capacity discrepancy between modalities, and powerful autoregressive decoders (Bai et al., 2024; Yin et al., 2023). Our work is the first to interpret hallucination in EEG-to-text through the lens of posterior collapse, and we respond on two fronts: effective EEG representation learning as well as quantitative semantic evaluation.

**Brain-model representation alignment.** Aligning brain activity with multimodal representations from pretrained models has provided important insights into the hierarchical structure of human perception. In vision, deep convolutional neural networks (CNNs) exhibit layer-wise correspondence with the primate visual cortex, where deeper layers consistently make better predictions of responses in higher cortical areas (Yamins et al., 2014; Cadieu et al., 2014; Cichy et al., 2016; Seeliger et al., 2018). This pattern extends to recent comparisons between vision models, evaluated by their alignment with non-invasive magnetoencephalography (MEG) signals: representations of self-supervised and contrastive learning models can be more accurately retrieved than those of classification models or hand-crafted features (Benchetrit et al., 2024). In speech, evidence also supports the hierarchical organization during auditory language processing (Caucheteux et al., 2023), while recent research has shown that self-supervised representations significantly outperform acoustic features in MEG/EEG-to-speech retrieval (Défossez et al., 2023). Notably, in language processing, multiple studies have found that middle layers of large language models (LLMs) than the earliest or latest layers better predict brain responses during natural language reading and listening (Schrimpf et al., 2021; Antonello et al., 2021; Jain & Huth, 2018; Toneva & Wehbe, 2019), an effect attributed to the representational generality of the middle layers (Antonello & Huth, 2024), as evidenced by superior transfer performance across downstream tasks (Skean et al., 2025). These converging findings suggest that high-level, abstract model representations align more closely with mental representations captured in non-invasive signals. Our method builds on this principle by explicitly aligning EEG representations with the latent space of a pretrained encoder-decoder LM, in contrast to prior work that relies on word-level alignment with embedding layers (Wang & Ji, 2022; Duan et al., 2023).

# 3 METHOD

## 3.1 PRELIMINARIES

**ZuCo dataset.** We use the publicly available ZuCo dataset (Hollenstein et al., 2018; 2019) as a motivational benchmark for our framework. ZuCo provides 128-channel EEG recordings collected during English sentence reading tasks, covering both normal reading (NR; passive reading) and task-specific reading (TSR; active reading with comprehension questions). It contains over 22K sentence-level EEG-text pairs, with categorical annotations available for all TSR samples and a subset of NR samples. Notably, ZuCo features two key advantages on EEG-to-text research: (1) representative data heterogeneity across reading paradigms, corpora, sessions, and subjects—posing a persistent challenge for training generalizable models; and (2) its inclusion of corpora such as SST (Socher et al., 2013) and Wiki (Culotta et al., 2006), which are widely used to benchmark language models on sentiment analysis and relation extraction, respectively—enabling seamless integration with pretrained LMs for both supervision and evaluation. Together, these properties establish ZuCo as a strong prototypical setting for collecting semantically evaluable large-scale datasets, and motivate our scalable and modular design in GLIM. Additional details on data statistics, preprocessing, and split are provided in Appendix A.

**Problem formulation.** We frame the EEG-to-text decoding as a semantic summarization task and aim to train a model that generalizes across domains while supporting quantitative semantic evaluation. Formally, given a set of sentence-level EEG time series $\{X_i \in \mathbb{R}^{L_t \times D_c}\}$ recorded while subjects read stimulus texts $\{Y_i\}$, where $L_t$ and $D_c$ denote the number of time points and EEG channels respectively, our training objective is to learn informative EEG representations that capture the core semantics of stimuli. To further improve generalizability and mitigate data scarcity, each training sample is accompanied by a domain-specific prompt $p_i$ and a set of $K$ text variants $\{Y_i^j \mid j = 1, 2, ..., K\}$ paraphrased from $Y_i$. At inference time, GLIM generates coherent sentences directly from EEG signals and domain prompts without teacher forcing—mediated through the learned EEG representations (see Fig. 2).

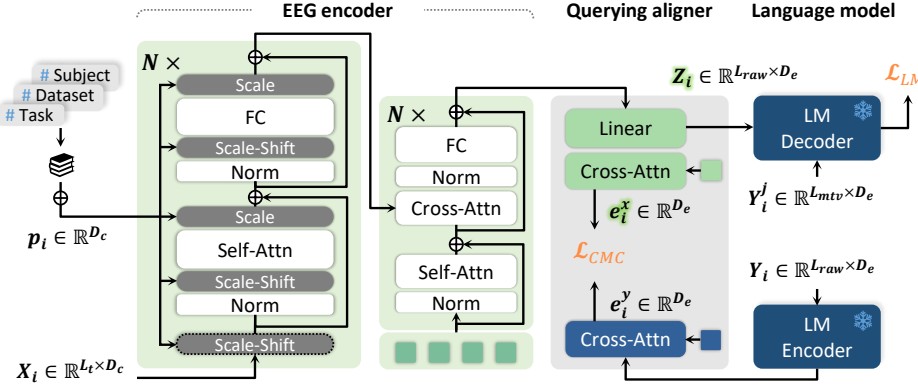

Figure 2: **Architecture and training objective of GLIM.** It consists of three modules: a domain-adaptive EEG encoder, a pretrained encoder-decoder language model (LM), and a cross-modal querying aligner. We train the EEG encoder and the querying aligner to align EEG representations with the latent space of the frozen LM. There are two forms of EEG representations: (1) a token-level sequence representation $Z_i$, used to generate sentences by conditioning the LM decoder; and (2) a global embedding $e_i^x$, enabling EEG-text retrieval and zero-shot semantic classification.

## 3.2 EEG ENCODER

**Temporal downsampling with cross-attention.** The EEG encoder transforms heterogeneous EEG time series into compact latent representations that match the length of sentence representations in the language model. As shown in Figure 2, it adopts a transformer-based encoder-decoder architecture. Inspired by Q-former (Li et al., 2023a), we introduce a fixed set of learnable queries that

attend to the EEG time series through cross-attention mechanism, enabling automatic and adaptive temporal downsampling. This design not only allows the encoder to capture the intrinsic temporal dependencies across two simultaneously acquired modalities, but also reduces the computational cost associated with long input sequence.

**Domain prompt injection.** As prompt injection has proven effective for improving joint training across heterogeneous datasets (Wu et al., 2024), we incorporate adapter modules to adapt the EEG encoder to multiple domains. Beyond aligning with the grouping in the ZuCo dataset, we construct three prompt-indexing dictionaries that represent common experimental conditions in natural reading tasks, characterized by *subject*, *dataset* and *task*. These factors respectively capture: (1) inter-individual variability in brain structure and function (Jayaram et al., 2016; Wei et al., 2021; da Silva Castanheira et al., 2021), (2) cross-session differences in hardware or setup (Hollenstein et al., 2019), and (3) behavioral differences between passive and task-driven reading paradigms (see Appendix. A for more details). Since these factors modulate the spatiotemporal patterns of EEG signals, we follow the adapter design from DiT (Peebles & Xie, 2023) to perform scale-shift normalization on most hidden layers in encoder-side blocks. Additionally, we add a normalization layer at the top of each encoder block to further provide temporal adaption, while the others address spatial (channel) variation. Prompt dropout (Peebles & Xie, 2023) is also applied to support inference on unseen domains, via replacing each of the three prompts with an `[UNKNOWN]` token under certain probabilities during training.

### 3.3 LANGUAGE MODEL

**Pretrained encoder-decoder model.** We integrate a frozen encoder-decoder language model, specifically Flan-T5 (Chung et al., 2024), which provides a structured latent space for both representation alignment and sentence generation. Compared to decoder-only models, encoder-decoder LMs are pretrained not only with autoregressive language modeling but also with masked language modeling (Lewis, 2019; Raffel et al., 2020), allowing them to encode sentence-level semantic representations that are more robust and interpretable. Furthermore, the Flan-T5 model is instruction-tuned on a wide range of natural language tasks, enabling it to perform semantic supervision and evaluation in our framework without additional finetuning. This setup supports the use of sentence embeddings for EEG alignment and prompt-based zero-shot classification for downstream evaluation.

**Multiple text variants.** As evidence shows that in text summarization tasks, constructing diverse target texts can effectively improve generation performance and reduce overfitting (Loem et al., 2022), we adopt a similar strategy to enhance the robustness of EEG-to-text decoding under limited data. Specifically, for each stimulus sentence, we generate multiple text variants (MTVs) that preserve core semantics while varying in surface form, encouraging the model to focus on abstract meaning rather than lexical patterns during autoregressive language modeling. We construct the variants using a large language model prompted with tailored rewriting instructions per sentence, covering diverse syntactic and lexical structures while maintaining semantic fidelity. Full rewriting instructions, prompt templates and semantic control strategies are provided in Appendix B.

### 3.4 QUERYING ALIGNER

To align the two modalities in a shared semantic space, we introduce a lightweight querying aligner (Q-aligner) composed of a linear projection layer and a cross-attention module with a learnable query token. The sentence embedding of each raw stimulus text $e_i^y$ is extracted by placing an instance of the Q-aligner after the LM encoder, omitting the projection layer. Another full instance is placed after the EEG encoder to derive both the sequence representation $Z_i$ and global embedding $e_i^x$. These representations jointly support both text generation and semantic evaluation. Notably, the Q-aligner plays a central role in our modular design: it enables seamless integration between the EEG encoder and any frozen language model. By providing a common semantic interface—projecting EEG signals from channel space and querying sentence representations from LM latent space—the Q-aligner supports flexible encoder substitution and parameter-efficient training.

## 3.5 TRAINING OBJECTIVE

We jointly train GLIM with two complementary objectives: an autoregressive language modeling loss essential for coherent sentence generation, and a cross-modal contrastive loss that further aligns EEG-text representations at the embedding level.

**Autoregressive language modeling.** For each EEG input, the language modeling is performed on multiple text variants. Given the sequence representation $Z_i$ derived from $X_i$ and $p_i$, the paraphrased variants provide complementary supervision on high-level, core semantics and guide the model to generate coherent sentences conditioned on $Z_i$. This can be formulated by:

$$\mathcal{L}_{LM} = -\frac{1}{NK} \sum_{i=1}^{N} \sum_{j=1}^{K} \log P(\hat{Y}_i^j | Z_i) \tag{1}$$

Where $N$ is the number of original training samples, $K$ is the number of variants per sample, and $\hat{Y}_i^j$ is the generated text under teacher forcing of its ground truth $Y_i^j$.

**Cross-modal contrastive learning.** Since pairing a powerful autoregressive decoder with noisy input is known to be prone to posterior collapse (Goyal et al., 2017), we introduce a contrastive learning objective to mitigate this imbalance. Following CLIP (Radford et al., 2021), we apply this objective over EEG-text embedding pairs in each training batch, maximizing the distance of non-matching pairs:

$$\mathcal{L}_{CMC} = -\frac{1}{2B} \sum_{i=1}^{B} \left( \log \frac{\exp(\theta(e_i^x, e_i^y))}{\sum_{j=1}^{B} \exp(\theta(e_i^x, e_j^y))} + \log \frac{\exp(\theta(e_i^x, e_i^y))}{\sum_{k=1}^{B} \exp(\theta(e_k^x, e_i^y))} \right) \tag{2}$$

Where $\theta$ denotes cosine similarity and $B$ is the batch size. This objective helps the model distinguish subtle differences between closely related sentences—particularly important under limited textual diversity.

**Total training objective.** Overall, the total loss function combines above two objectives with a weighted sum:

$$\mathcal{L}_{total} = \lambda \mathcal{L}_{LM} + (1 - \lambda) \mathcal{L}_{CMC} \tag{3}$$

Where $\lambda \in [0, 1]$. This end-to-end training objective explicitly encourages the model to learn informative EEG representations against posterior collapse, while supporting both coherent sentence generation and quantitative evaluation at latent space.

## 4 EXPERIMENT

We evaluate GLIM on the ZuCo dataset, following prior EEG-to-text work while adopting a stricter 8:1:1 split to ensure that no same sentence appears in both training and test sets. In contrast to previous approaches that merely rely on surface-level text similarity, we combine three complementary evaluation metrics to comprehensively inspect the decoded semantics in both generated texts and EEG representations. We therefore set two baselines for comparison: the *EEG2text* model reproduced by Jo et al. (2024) and a random baseline (i.e., the chance-level accuracy of retrieval and classification, denoted as the $\mathcal{U}$ baseline).

In this section, we first introduce the evaluation metrics and the critical "noise input test", then present comparative results and ablations to validate GLIM's reliable semantic decoding capability and design rationality. We further examine whether focusing on informative EEG representations learning and their semantic evaluation contributes to improving the faithfulness of EEG-grounded text generation. Finally, we assess GLIM's ability to scale across heterogeneous data domains by evaluating its joint training and transfer performance.

### 4.1 EVALUATION PROTOCOLS

**Generation.** In contrast to prior work, GLIM directly generates natural sentence from EEG input without teacher forcing, which can be seamlessly implemented by conditioning the LM's decoder on the learned EEG sequence representation with LM's default generation settings (e.g., beam search). Since our model focuses on semantic fidelity rather than word-level matching, we use BLEU-1 and BLEU-2 scores calculated with multiple references (i.e., the multiple text variants, denoted

by *@MTV*) to measure the semantic precision of generated content. Additionally, we report the ROUGE-1-Recall that calculated against the raw stimulus text (*@RAW*), providing comparison with the baseline model while reflecting the feasibility of finely reconstructing stimulus texts.

**Retrieval.** To evaluate how well the learned EEG representations capture the subtle differences between similar sentences, we compute the EEG-text retrieval accuracy (top-1 and top-5) by retrieving matched sentences from the EEG embeddings within each subgroup—the smaller evaluation batch (of size 24) grouped by reading task, subject, dataset as well as the corpus source.

**Zero-shot classification.** To evaluate the core semantic capturing in EEG representations, we perform zero-shot classifications on sentiment, relation type, and corpus source, where each task is conducted on different subsets depending on annotation availability (details in Appendix A). The implementation follows CLIP (Radford et al., 2021), where we directly use the integrated LM's encoder (and Q-aligner) to obtain label embeddings and compute their cosine similarities with each EEG embedding, deriving the classification probabilities. Additionally, we also implement an LLM-assisted classification to assess the semantic fidelity of generated texts, as detailed in Section 4.3.

**Noise input test.** Following Jo et al. (2024), we conduct the "noise input test" for each run (denoted by $\mathcal{N}_{in}$) to examine whether the decoding truly rely on EEG inputs, eliminating any other confounder, such as the domain prompt inputs and the language model prior. This is approached by simply replacing each EEG input with Gaussian noise at test time.

## 4.2 Evaluating Semantic Fidelity and Representation Alignment

Table 1: Performance comparison and ablation studies. **Generation** metrics are averaged over all test samples; **Retrieval** is computed as the average across subgroups of 24 sentences; **Classification** accuracies are computed on annotation-specific test subsets. †: Reported from a different data split with potential train-test text overlap; used here for approximate reference. ∗: BLEU scores computed against raw stimulus text, not our multiple text variants.

| Model | Generation | | | Retrieval | | Classification | | |
|---|---|---|---|---|---|---|---|---|
| | BLEU1 @*MTV* | BLEU2 @*MTV* | ROUGE1 @*RAW* | ACC-1 | ACC-5 | ACC-1 Sentiment | ACC-1 Relation | ACC Corpus |
| EEG2Text | $0.1675^{\dagger,*}$ | $0.0615^{\dagger,*}$ | $0.1527^{\dagger}$ | - | - | - | - | - |
| EEG2Text ($\mathcal{N}_{in}$) | $0.1570^{\dagger,*}$ | $0.0544^{\dagger,*}$ | $0.1384^{\dagger}$ | - | - | - | - | - |
| $\mathcal{U}$ baseline | - | - | - | 0.0417 | 0.2083 | 0.3333 | 0.1111 | 0.5000 |
| **Ours** | **0.2604** | **0.1056** | 0.1227 | **0.0815** | **0.3510** | **0.4269** | **0.3245** | **0.9348** |
| **Ours** ($\mathcal{N}_{in}$) | 0.1824 | 0.0451 | 0.1111 | 0.0367 | 0.2070 | 0.3573 | 0.1449 | 0.6273 |
| w/o $\mathcal{L}_{LM}$ | 0.0000 | 0.0000 | 0.0003 | **0.0734** | **0.2939** | 0.2901 | 0.2122 | 0.4135 |
| w/o $\mathcal{L}_{LM}$ ($\mathcal{N}_{in}$) | 0.0000 | 0.0000 | 0.0006 | 0.0403 | 0.2088 | 0.2686 | 0.1806 | 0.2120 |
| w/o $\mathcal{L}_{CMC}$ | 0.1833 | 0.0511 | **0.1238** | 0.0408 | 0.2120 | **0.4341** | 0.0745 | **0.8211** |
| w/o $\mathcal{L}_{CMC}$ ($\mathcal{N}_{in}$) | 0.1769 | 0.0424 | 0.1014 | 0.0412 | 0.2079 | 0.4341 | 0.0745 | 0.8121 |
| w/o MTV | **0.2064** | **0.0518** | **0.1258** | 0.0571 | 0.2246 | 0.2758 | **0.2449** | 0.7237 |
| w/o MTV ($\mathcal{N}_{in}$) | 0.1585 | 0.0327 | 0.1369 | 0.0430 | 0.2056 | 0.2829 | 0.0265 | 0.6372 |

**GLIM exhibits reliable EEG-grounded decoding performance.** We first compare GLIM with *EEG2Text* and the random baseline. As shown in Table 1, GLIM significantly outperforms these baselines across generation, retrieval, and classification tasks. Notably, the high zero-shot classification accuracies on abstract semantic categories—such as sentiment, relation types, and corpus sources—demonstrate its strong capability in decoding EEG-grounded semantics. To our knowledge, this is the first demonstration of such zero-shot evaluation in EEG-based generative decoding.

**Each training objective contributes uniquely to decoding fidelity.** The ablation studies further validate the necessity of our two training objectives. Removing the language modeling loss $\mathcal{L}_{LM}$ results in meaningless generation outputs and degenerate classification accuracy, underscoring its role in enabling generation and learning meaningful representations. In contrast, removing the contrastive loss $\mathcal{L}_{CMC}$ leads to a sharp decline in retrieval accuracy and minimal performance gap under the noise input condition, suggesting the model is prone to posterior collapse (by overfitting to text priors or learning spurious prompt-label correlations) without this regularization.

**Multiple text variants improve semantic robustness.** Finally, we observe that the use of MTV significantly improves generation and representation quality. Ablating MTV leads to consistent drops in performance across all metrics (except for ROUGE1@*RAW*, which favors surface-form matching) and diminished EEG-noise gap. This confirms that MTV robustly guide the model to extract core semantics and avoid overfitting to limited linguistic patterns, thus effectively mitigating posterior collapse.

### 4.3 INSPECTING SEMANTIC CONSISTENCY IN GENERATED TEXTS

While the previous section verifies that GLIM learns informative EEG representations, a key question remains: *Do the generated sentences themselves preserve the decoded semantics?* To answer this, we evaluate semantic consistency between EEG embeddings, generated texts, and embeddings of generated texts. We adopt two complementary evaluation strategies. First, we perform CLIP-like zero-shot classification on EEG and text embeddings. Second, we apply an LLM-assisted classification that prompts an advanced LLM to predict semantic categories of the generated sentences themselves (the same LLM we use to generate text variants). For reference, we also include the performance on raw stimulus texts and their embeddings to establish soft upper bounds. Details about the generated text samples are demonstrated in Appendix C.

Table 2: Semantic classification accuracies across different outputs. **LLM-assisted** classification uses direct prompt-based inference. **CLIP-like** method computes cosine similarity over embeddings, and its result on EEG embedding (first row) corresponds to out best model in Table 1.

| Output | Method | ACC-1 Sentiment | ACC-1 Relation | ACC-3 Relation | ACC Corpus |
|---|---|---|---|---|---|
| **EEG embedding** | **CLIP-like** | **0.4269** | **0.3245** | **0.5714** | **0.9348** |
| Gen text | LLM-assisted | 0.3957 | 0.0724 | 0.5633 | 0.9216 |
| Gen text embedding | CLIP-like | 0.3957 | 0.2071 | 0.4326 | 0.8736 |
| Raw text | LLM-assisted | **0.7338** | 0.0969 | **0.7551** | 0.8614 |
| Raw text embedding | CLIP-like | **0.4556** | 0.2530 | 0.4969 | 0.9185 |

**Generated texts consistently reflect EEG-derived semantics.** Table 2 shows that GLIM's generated sentences achieve comparable classification accuracies to those of EEG embeddings. This consistency supports our key methodological emphasis—*learning interpretable EEG representations leads to semantically faithful generation*. While embeddings of generated texts yield slightly lower accuracy, the drop is expected due to the LM encoder's lossy compression.

**Complementary evaluation reveals supervision strength.** Although the classification accuracies of raw stimulus texts and their embeddings can be considered intuitive upper bounds for semantic evaluation and supervision, they exhibit notable limitations. The former suffers from label ambiguity and LLM prior bias—particularly in relation classification, where many text samples lack a clear one-to-one label correspondence, leading to sharp drops in top-1 accuracy. The latter fails to fully capture sentence-level semantics, indicating that the LM encoder alone provides insufficient supervision. In contrast, GLIM achieves consistently high accuracy across both generated texts and EEG embeddings, even surpassing these baselines. These results underscore the importance of combining robust semantic supervision with complementary evaluation protocols—core components of GLIM's design for effective and reliable decoding.

### 4.4 ASSESSING GENERALIZABILITY ACROSS HETEROGENEOUS DOMAINS

As introduced in Section 3.2, we apply prompt dropout during training our best model, with dropout probabilities of $\{0, 0.1, 0.1\}$ for *task*, *dataset* and *subject*, respectively. This section first evaluates GLIM's generalization to unknown datasets and subjects by disabling $\{d, s\}$ prompts at test time. In addition, we train three ablated models, each with specific prompts disabled during training, to quantify the contribution of each domain prompt.

**GLIM generalizes well to unspecified subjects and datasets.** As shown in Table 3, our best model maintains high performance even when *dataset* and *subject* prompts are disabled at test time. This elucidates that the model does not rely on the identification of prompt-specific information

Table 3: Ablation study of domain prompt injection. **{t,d,s}** indicates the prompt types activated during training or test. The first row corresponds to our best model (same in Table 1).

| Prompts | | Generation | | | Retrieval | | Classification | | |
|---|---|---|---|---|---|---|---|---|---|
| **Train** | **Test** | **BLEU1** @*MTV* | **BLEU2** @*MTV* | **ROUGE1** @*RAW* | **ACC-1** | **ACC-5** | **ACC-1** Sentiment | **ACC-1** Relation | **ACC** Corpus |
| {t,d,s} | {t,d,s} | **0.2604** | **0.1056** | **0.1227** | 0.0815 | **0.3510** | **0.4269** | **0.3245** | **0.9348** |
| {t,d,s} | {t} | **0.2682** | **0.1091** | **0.1282** | 0.0802 | 0.3401 | **0.4244** | 0.3112 | **0.9076** |
| Ø | Ø | 0.2223 | 0.0646 | 0.1142 | **0.0973** | **0.3687** | 0.2902 | 0.2694 | 0.6341 |
| {t} | {t} | 0.2434 | 0.0936 | 0.1084 | **0.0865** | 0.3053 | 0.3142 | **0.3694** | 0.4592 |
| {d,s} | {d,s} | 0.2056 | 0.0594 | 0.1085 | 0.0770 | 0.3152 | 0.3309 | 0.3194 | 0.6223 |

to express domain-dependent priors. Instead, its backbone effectively learns shared core semantics across heterogeneous EEG data, while the adapter modules provide spatiotemporal adaptation independently. Detailed subgroup performance comparisons are provided in Appendix D.

**Task prompt captures paradigm-induced brain variability.** Disabling prompts during training consistently reduces performance, confirming that all three domain prompts contribute to robust joint training. Among them, the *task* prompt has the most significant impact. This supports our hypothesis that normal reading and task-specific reading elicit systematically different brain states, which manifest as distinct spatiotemporal patterns in EEG time series. Incorporating task-type information helps the model adapt to these differences and improves overall generalization.

## 5 DISCUSSION

**Limitations.** While GLIM effectively enhances the semantic faithfulness of EEG-to-text decoding, several limitations remain. First, our latent-space alignment strategy primarily targets the intermediate representations of a frozen pretrained language model, without fully leveraging the LM's text-to-text capabilities. Although this design mitigates posterior collapse and improves interpretability, the exclusion of semantic priors in encoder's upstream representations may limit the upper bound of decoding performance and introduce supervision biases. Second, when fine-grained lexical details are partially encoded in EEG signals, the use of multiple paraphrased text variants (MTVs) may dilute or obscure such signals. While all variants emphasize the shared core semantics, they may inconsistently suppress secondary meanings—potentially hindering the model's ability to reconstruct more specific linguistic content.

**Future work.** As demonstrated in our experiments, GLIM establishes a scalable and interpretable prototype for future large-scale EEG-to-text pretraining. Moving forward, we aim to extend this work in two directions: (1) enhancing end-to-end semantic decoding accuracy, and (2) advancing toward practical non-invasive language BCI systems. The former involves exploring improved cross-modal alignment strategies, integrating stronger language models, and scaling up both model capacity and training data. The latter builds on GLIM's ability to produce coherent, semantically grounded sentences and may benefit from post-generation policies—such as human feedback or reward modeling—to further improve usability in real-world applications.

**Conclusion.** We clarify posterior collapse as the root cause of hallucination in current EEG-to-text methods and introduce GLIM to emphasize informative, interpretable representation learning across heterogeneous domains. Our work takes a concrete step toward reliable and scalable modeling and evaluation, laying the foundation for future scaling laws in generative brain decoding.

### REPRODUCIBILITY STATEMENT

All source code including complete data preprocessing and splitting scripts (see Appendix A for more details) is anonymized and included in supplementary material. We commit to release the full codebase (along with model checkpoints) public upon publication decision, supporting open research and trustworthy benchmarking in EEG-to-text decoding.

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

# Appendix

## A DATASET

Our modeling and evaluation protocols are tightly integrated with the experimental design of the ZuCo dataset, which features notable domain heterogeneity, language model-compatible corpora, and semantically annotated reading tasks. We believe ZuCo offers a prototypical paradigm for future large-scale EEG-text datasets collected during natural reading. This section first introduces the dataset and group statistics, followed by our unified EEG preprocessing strategy, strict data splitting protocol, and suggestions for future data collection.

### A.1 ZUCO OVERVIEW

The complete ZuCo dataset (comprising ZuCo 1.0 (Hollenstein et al., 2018) and 2.0 (Hollenstein et al., 2019)) contains 128-channel EEG recordings sampled at 500Hz during English sentence reading. The texts are drawn from the Stanford Sentiment Treebank (SST), annotated with sentiment categories (*neutral*, *negative*, *positive*), and from the Wikipedia relation extraction corpus (Wiki), labeled with relation types such as *awarding*, *education*, *employment*, *foundation*, *job title*, *nationality*, *political affiliation*, *visit* and *marriage*.

ZuCo features two reading paradigms: normal reading (NR) and task-specific reading (TSR). NR sessions involve passive reading of corpus-specific sentences with occasional control questions. TSR sessions are centered on a specific relation type, with most sentences accompanied by question-answering (QA) tasks, ensuring semantic comprehension and mental grounding.

Table 4: Group-level statistics of the ZuCo dataset.

| Group | Dataset | Reading paradigm | Corpus | Label available | QA | Subject num | Sentence num | Sentence length | Reading time |
|---|---|---|---|---|---|---|---|---|---|
| I | ZuCo1 | NR | SST | Sentiment | - | 12 | 400 | 17.7 | 5.5 s |
| II | ZuCo1 | NR | Wiki | - | - | 12 | 300 | 21.3 | 7.2 s |
| III | ZuCo1 | TSR | Wiki | Relation | ✓ | 12 | 407 | 20.1 | **4.2 s** |
| IV | ZuCo2 | NR | Wiki | - | - | 18 | 349 | 19.6 | 5.8 s |
| V | ZuCo2 | TSR | Wiki | Relation | ✓ | 18 | 390 | 21.3 | **4.8 s** |

### A.2 DOMAIN SPLIT AND EVALUATION GROUPS

Table 4 highlights ZuCo's domain variability. To enable effective joint training, GLIM uses prompt-based domain adaptation across three factors: reading paradigm (*task*), dataset version (*dataset*), and subject identity (*subject*). Among them, the *task* prompt is particularly important, motivated by the distinct cognitive processes in NR versus TSR, reflected in the consistent differences in reading time (Hollenstein et al., 2018).

For evaluation, we further split test data by corpus (SST or Wiki) to allow fine-grained analysis. In classification tasks, metrics are averaged across the applicable subgroups. Since relation labels are only available for TSR-Wiki samples, relation classification is restricted to that subset. Corpus-level annotations (movie review vs. biography) were manually added for all test samples.

### A.3 EEG PREPROCESSING

To preserve information and facilitate scaling, we apply minimal preprocessing. Specifically: (1) EEG signals are downsampled from 500Hz to 128Hz and zero-padded to 1280 time points (10 seconds); (2) Channels are padded from 104 to 128.[1] This produces uniformly shaped EEG sequences, enabling efficient training (e.g., 128 being a GPU-efficient multiple of 8) and seamless integration of new datasets.

---

[1] The 105th channel contains all NaNs across samples and is excluded in our processing. Surprisingly, this issue is not documented in prior studies despite widespread use of this dataset.

### A.4 DATA SPLIT

To prevent data leakage, we split based on unique stimulus texts, ensuring that no text appears in more than one of the train/val/test sets. Given ZuCo's intentional overlap across subjects, paradigms and datasets,[2] we first collect all overlapping samples into the training set, then randomly sample from the remaining unique sentences (with a fixed seed) in a stratified manner. The final split is 17908/2200/2227 (approximately 8:1:1).

### A.5 RECOMMENDATIONS FOR LARGE-SCALE DATA COLLECTION

Building on GLIM and the ZuCo dataset, we recommend the following guidelines for future large-scale EEG-to-text datasets:

- Use text corpora aligned with language model downstream tasks;
- Include QA tasks to ensure semantic comprehension;
- Record comprehensive metadata (e.g., paradigm, device, language) to enable domain-aware modeling.

As wearable EEG devices improve (Zhang et al., 2023), data collection in natural reading—compared to typing (Lévy et al., 2025) or silent-speech paradigms (Nieto et al., 2022; Zhou et al., 2025)—remains low-cost and consistent, requiring only a screen and simple interface, making it ideal for broad adoption.

## B MULTIPLE TEXT VARIANTS

### B.1 CONSTRUCTION

To mitigate overfitting and guide the model toward high-level semantic alignment, we construct multiple paraphrased variants for each raw stimulus text. Specifically, we use *Llama3.1-70B-Instruct* (Dubey et al., 2024) to generate six variants following three rewriting rules—lexical simplification, semantic clarity, and syntactic simplification (two per rule)—each aimed at emphasizing distinct aspects in language use. Additionally, we use the integrated LM (*Flan-T5-Large*) to produce two simpler variants using natural language prompts ("To English: ..." and "Summarize: ..."). Table 5 summarizes the variant types and instructions.

Table 5: Overview of variant types and corresponding rewriting rules.

| Variant type | Num | Rephrasing instruction / Prefix |
|---|---|---|
| Lexical simplification (LS) | 2 | ..., focusing on the choice of words used in the sentence, such as using simpler and more common words, avoiding jargon and technical terms. |
| Semantic clarity (SC) | 2 | ..., ensuring the meaning of sentence is clear and unambiguous, such as limiting the use of pronouns, completing the missing subject or object. |
| Syntax simplification (SS) | 2 | ..., altering the structure of sentence to make it easier to understand, such as using active voice, reducing clauses to phrases. |
| General rewritten (GR) | 1 | To English: ... |
| General simplification (GS) | 1 | Summarize: ... |

To ensure the preservation of core semantics, we provide each variant generation prompt with supplementary label information (e.g., sentiment categories for NR-SST; candidate/true relation types

---

[2]Prior studies (Wang & Ji, 2022; Jo et al., 2024) did not consider the latter two overlapping conditions in their subject-stratified splits; see their public codes: https://github.com/MikeWangWZHL; https://github.com/NeuSpeech.

for NR-Wiki/TSR-Wiki). These variants not only introduce surface-level linguistic diversity but also serve different supervision roles. In particular, the "General Rewritten" variant—generated by the integrated LM using the same prompt as training—is treated as a reference target for modeling the LM's text-to-text prior.

### B.2 ANALYZING VARIANT EFFECTIVENESS

In Section 4.2, we show that the use of MTV enhances generation and representation quality. Here, we examine how different variant types individually contribute to performance. We compute BLEU-1 and ROUGE-1 recall scores between model-generated texts and each variant, including comparisons against noise input baselines. The results are visualized in Figure 3, alongside pairwise significance tests using Welch's $t$-test.

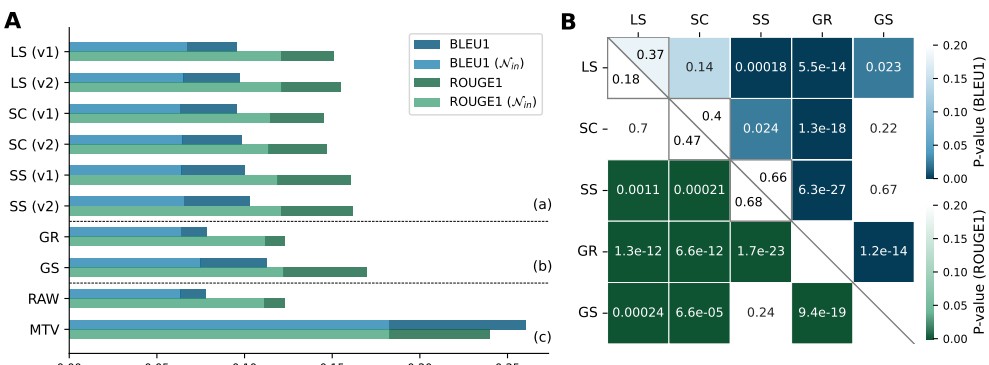

Figure 3: **(A) Average generation scores per variant type.** Light bars denote average BLEU-1 and ROUGE-1 scores under noise input tests ($\mathcal{N}_{in}$); while dark bars show *absolute improvements* over the averaged $\mathcal{N}_{in}$ scores (i.e., $Score - Score_{\mathcal{N}_{in}}$). **(a)** Six LLM-generated variant types. **(b)** Two types generated by the integrated LM. **(c)** Baseline references calculated with raw stimulus texts or with all 8 variants (i.e., our main results). **(B) Heatmap of pairwise $p$-values for variant comparisons.** The diagonal blocks represent comparisons within the same variant type, while the other blocks illustrate the inter-type comparisons. The $p$-values are calculated using the *absolute-improvement scores*; a value of $p < 0.05$ indicates a significant difference.

**Simplified variants support EEG-grounded generation.** We observe that variants generated under the same rewriting rule yield consistent results, confirming the distinct contribution of each variant type. Among the three LLM-generated types, *syntactic simplification* variants consistently gain most significant *absolute-improvement scores* in both BLEU-1 and ROUGE-1. This suggests that simplifying structural complexity helps the model better align with the latent semantic patterns encoded in EEG signals—possibly because this simplification method better simulates the top-down sentence processing of human brain (Mirault et al., 2018; Milledge et al., 2023)—or better matches the preferred text-to-text modeling of the integrated language model (Papadimitriou et al., 2022; Chen et al., 2024).

**LM prior alone is insufficient for semantic alignment.** Although the *general rewritten* variant type is directly generated by the same integrated LM and shares the same training prompt ("To English: ..."), it results in the lowest semantic overlap with model outputs. This indicates that such variants, while fluently phrased, are less effective for guiding EEG-grounded semantic decoding—highlighting the limited utility of relying solely on LM priors without simplification. On the other hand, the *general simplification* variant achieves relatively high scores even under noise input, suggesting that matching the LM prior helps model fluency but does not contribute substantially to semantic alignment.

**Variant diversity promotes semantic robustness.** Taken together, these findings show that no single variant type dominates the learning process. Instead, the collective linguistic diversity provided by MTV enables the model to abstract away from surface-level word forms and focus on core semantic content. This abstraction is critical for learning shared representations that are robust

across subjects and contexts. Compared to corresponding high semantic accuracies, the low word-overlap rate between generated texts and the raw stimulus texts further confirm that GLIM learns to decode high-level semantics rather than memorize input words.

## C  GENERATED SAMPLES

To qualitatively assess GLIM's generation ability, we present representative samples in Figure 4. Despite the frequent presence of hallucinations and stylistic repetition, the generated texts are generally fluent, grammatically correct, and express core semantic content with diverse paraphrasing. Two findings are particularly noteworthy. First, across subjects and reading conditions, corpus-level distinctions (e.g., movie reviews vs. biographies) are consistently presented, while the accuracy of sentiment and relation expressions varies—mirroring the quantitative metrics. This disparity reflects task-driven semantic engagement: sentiment labels in the NR paradigm are only sporadically addressed through control questions, whereas in TSR, each sentence is explicitly paired with a relation-type query. The relatively low sentiment decoding accuracy thus likely stems from limited neural encoding, rather than model failure. Second, the semantically anchored generative diversity supports our central hypothesis: *abstract, high-level semantics are more robustly represented in EEG signals than surface-level lexical forms.* This aligns with our model design, which emphasizes high-level semantic alignment over word-forced language memorization.

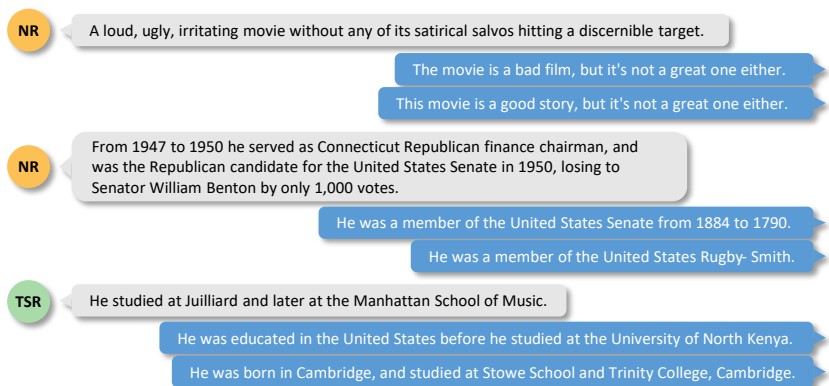

Figure 4: **Representative examples of generated texts.** The three groups correspond to NR-SST, NR-Wiki, and TSR-Wiki, each showing the raw stimulus and two generated texts from different subjects. We observe: (1) corpus distinctions are regularly captured (movie reviews in SST vs. personal bios in Wiki); (2) relation types are expressed diversely, especially in the TSR group (e.g., the "education" label is paraphrased as "educated" and "studied"); (3) hallucinations mainly involve contradictory logic and irrelevant content; and (4) repetitive sentence patterns appear but differ across corpus topics ("The movie..." vs. "He was...").

To further demonstrate the generation quality and semantic grounding of our model, we provide the complete set of texts generated by GLIM at the following share link: https://wandb.ai, which contains no identifying information and support anonymous browsing without login required. These examples cover all generated samples along with corresponding stimulus texts, and text variants. Moreover, we also provide the outputs from (1) the noise input test (corresponding to the 5th row in Table 1) and (2) the prompt-free test (2nd row in Table 3). These allow readers to fully inspect the semantic consistency between EEG representations and generated texts.

## D  CROSS-DOMAIN COMPARISON

To further assess GLIM's robustness under domain heterogeneity, we compare model performance across the five groups in ZuCo dataset (as in Table 4), each representing a unique combination of dataset version, reading paradigm and corpus source. As shown in Figure 5, each point represents the average performance of a single subject within a specific group, providing a subject-wise view of metric variation under controlled experimental conditions.

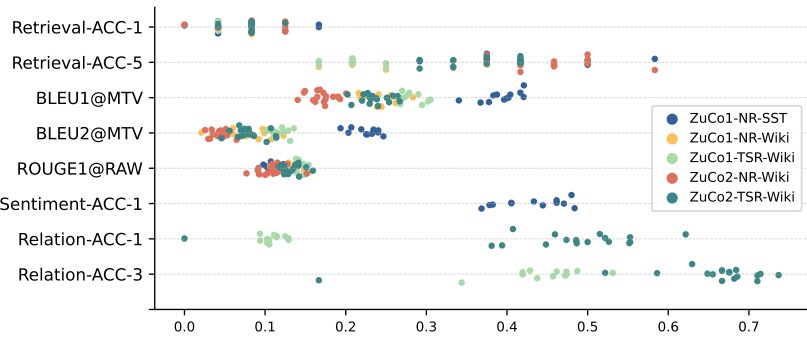

Figure 5: Performance comparison across different groups. The five groups correspond to various experimental conditions in the ZuCo dataset, with each dot representing the average metric for each subject.

**Reading paradigm and task comprehension affect decoding accuracy.** Within the Wiki corpus, we observe that TSR consistently outperforms NR in generation metrics. This supports our hypothesis that active engagement through question-answering (QA) promotes stronger semantic grounding, as subjects in TSR are required to comprehend the relation type of each sentence. In contrast, NR only involves passive reading with sparse comprehension checks, leading to more variable or weaker semantic encoding. Additionally, when comparing the *relative improvement* ($\frac{ACC - ACC_{\mathcal{N}_{in}}}{ACC_{\mathcal{N}_{in}}}$) in classification accuracy between NR-SST and TSR-Wiki, the top-1 sentiment accuracy increases by 40.3% over the noise input test; while the top-1 relation accuracy gains 123.9% (as in Table 1). This discrepancy further highlights the importance of grounding the semantic activation with QA steps when collecting natural reading datasets.

**Cross-dataset differences reveal domain effects.** Comparing ZuCo1 and ZuCo2, we observe a slight trade-off: while ZuCo1 achieves higher generation metrics, ZuCo2 outperforms in zero-shot classification. This may reflect inter-dataset variability in recording quality, subject population, or experimental protocols, all of which are common sources of domain shift in neural data. Importantly, despite these differences, GLIM maintains strong and consistent performance across all groups, confirming its capacity to adapt to domain heterogeneity through prompt-injected joint training.

# E    IMPLEMENTATION DETAILS

We implemented GLIM using the *PyTorch* framework and organized training using *PyTorch-Lightning*. The pretrained language model was *Flan-T5-Large*, integrated via *HuggingFace Transformers* (Wolf, 2019). Our final model stacked 6 + 6 encoder-decoder blocks in EEG encode, with the entire model containing 802M parameters, of which only 18.8M (2.34%) were trainable. All training were conducted on 8 × NVIDIA RTX-4090D-24GB GPUs using Distributed Data Parallel (DDP) for 200 epochs, with a batch size of 64, taking approximately 7 hours per run.

The MTV-augmented training set included eight paraphrased text variants per stimulus, resulting 143K triplets of *EEG-stimulus-variant*. To support contrastive learning within batches, we performed random sampling over unique stimulus texts during training. For validation, we fixed the batches with a size of 24, matching the number of unique texts in all test subgroups. Global random seeds were fixed across trials and epochs to ensure reproducibility.

