# OpenReview forum: "Learning Interpretable Representations Leads to Semantically Faithful EEG-to-Text Generation"
_ICLR.cc/2026/Conference — Submitted to ICLR 2026_

### Official Review · Reviewer_pGpW · 2025-10-31

**Soundness:** 2
**Presentation:** 2
**Contribution:** 2
**Rating:** 2
**Confidence:** 4

**Summary:**

This paper addresses the hallucination problem in EEG-to-text generation by reframing it as a semantic summarization task rather than verbatim reconstruction. The authors identify posterior collapse as a core issue, where powerful language decoders ignore noisy EEG inputs and generate plausible but unfaithful text. To mitigate this, they propose GLIM (Generative Language Inspection Model), which learns interpretable EEG representations aligned with a frozen language model’s latent space.

In previous EEG-to-text work, even when the input contained noise, sentences with relatively high scores could still be generated. This "illusion" can be explained as posterior collapse. But my personal alternative explanation is that, for example, in the silent reading tasks represented by the Zuco dataset, the brain activity is relatively weak, which results in the semantic features being less prominent. Therefore, the original EEG data maintains a similar data distribution to the noise. Therefore, the original EEG data maintains a similar data distribution to the noise. This view can be verified by the loss curve. When inputting noise, the training curve exhibits a similar fluctuating trend to the input EEG.

In fact, MEG datasets that focus on auditory tasks (such as Libribrain [1], Armeni [2], etc.) are relatively more meaningful and are considered suitable for Brain-to-text applications.

[1] Özdogan, M., Landau, G., Elvers, G., Jayalath, D., Somaiya, P., Mantegna, F., ... & Jones, O. P. (2025). LibriBrain: Over 50 Hours of Within-Subject MEG to Improve Speech Decoding Methods at Scale. arXiv preprint arXiv:2506.02098.

[2] Armeni, K., Güçlü, U., van Gerven, M., & Schoffelen, J. M. (2022). A 10-hour within-participant magnetoencephalography narrative dataset to test models of language comprehension. Scientific Data, 9(1), 278.

**Strengths:**

Novel Problem Framing: Reframing EEG-to-text as semantic summarization is a key insight, aligning the task with the noisy and abstract nature of EEG signals.

Robust Evaluation: Introduces zero-shot classification and EEG-text retrieval as complementary metrics, moving beyond surface-level text similarity.

Modular Design: The Q-aligner enables plug-and-play integration with any frozen LM, enhancing scalability.

Empirical Rigor: Ablation studies and the noise input test provide strong evidence for the model’s reliance on EEG signals.

Practical Impact: Lays the groundwork for scalable EEG-to-text pretraining and non-invasive language BCIs.

**Weaknesses:**

1.Limited Baseline Comparison: Only compares against EEG2Text and a random baseline. Recent models like DeWave or Brain2Text are not included, limiting claims of state-of-the-art performance.

2.EEG Preprocessing Simplicity: The minimal preprocessing (downsampling, zero-padding) may overlook domain-specific EEG artifacts, potentially affecting generalizability.

3.MTVs Trade-off: While multiple text variants improve robustness, they may dilute fine-grained lexical signals partially encoded in EEG, as acknowledged by the authors.

4.Scalability Concerns: The model relies on frozen LMs (Flan-T5), which may limit adaptability to larger or more recent LMs (e.g., LLaMA 3).

5.Limited Cross-Domain Analysis: While the model handles ZuCo’s heterogeneity, generalization to other EEG datasets (e.g., silent speech and listening task) remains untested.

**Questions:**

1.Baseline Comparison: Why not compare with DeWave or Brain2Text? How would GLIM perform against these recent models?

2.EEG Preprocessing: Have you explored artifact removal (e.g., ICA, filtering) or frequency-based features to improve signal quality?

3.MTVs Impact: How do you ensure that paraphrased variants preserve subtle semantic nuances (e.g., negation, sarcasm) critical for sentiment?

4.Scalability: Could GLIM be extended to larger LMs (e.g., LLaMA 3) or decoder-only models? What are the computational trade-offs?

5.Cross-Domain Generalization: How would GLIM perform on silent speech EEG datasets (e.g., Nieto et al., 2022)? Would the semantic summarization framework still apply?

---

> ### Author Response · Authors · 2025-11-26
>
> We thank the reviewer for the feedback and particularly appreciate the **five strong strengths** listed above. However, we have observed several **factually incorrect premises** and **out-of-scope suggestions** in the review. We provide point-by-point clarifications below.
>
> ---
> ### **A1. EEG-Noise distribution and "Loss Curve" (Summary P2)**
>
> The reviewer claims: *"original EEG data maintains a similar data distribution to the noise... verified by the loss curve"*. We must correct this factual error:
>
> * **EEG≠Noise**: Extensive EEG studies and our empirical results directly contradict the claim that *"EEG shares a distribution with noise"*. As shown in our **noise input test (Table 1, Row 5)** , replacing EEG with random noise causes performance to drop drastically to chance levels. If the signal were merely noise distribution, this collapse would not occur.
> * **Unfounded Evidence:** Our mauscript **does not contain any training loss curves**. The reviewer appears to be citing visual evidence that does not exist.
>
> ---
> ### **A2. Modality/Task-Mismatched Datasets (Summary P3, W5, Q5)**
>
> Based on the "personal explanation" of EEG's limited signal quality, the reviewer suggests using MEG datasets like LibriBrain (Ozdogan et al., 2025) or Armeni et al. (2022). Besides, the reviewer requests the generalization analysis over distinct cognitive tasks, such as silent speech datasets (Nieto et al., 2022).
>
> We clarify that our work focuses on **EEG-based Reading** decoding. EEG and MEG differ fundamentally in **signal quality** and **practicality**, so do the reading and listening/speech tasks. These citations and requests are **out of scope**.
>
> ---
> ### **A3. Baseline Comparisons (W1, Q1)**
>
> The reviewer asks for comparisons with DeWave or "Brain2Text".
>
> * **DeWave:** As detailed in our **General Response**, DeWave does not provide reproducible code. Comparing against reported numbers from a methodologically different (teacher-forced) setup is invalid.
> * **"Brain2Text":** If the reviewer refers to the work "Brain-to-text decoding... via typing" (Lévy et al., 2025) cited in our references, we clarify that it decodes **motor commands (typing)**, not semantic thoughts from reading. If referring to a generic model name, we note that no reproducible open-source baseline matches our setting.
>
> ---
> ### **A4. Minimal Preprocessing (W2, Q2)**
>
> The reviewer critiques our "minimal preprocessing"of EEG.
>
> * **End-to-End Philosophy:** Modern deep learning for EEG (e.g., EEGNet, Conformer) consistently shows that **end-to-end learning** on raw/minimally processed data outperforms heavy manual feature engineering (artifact removal/ICA), as neural networks can learn to filter noise data-drivenly.
> * **Benchmark Utility & Scalability:** Minimal preprocessing ensures that our benchmark is easy to adopt and scale on large-scale, heterogenous datasets.
>
> ---
> ### **A5. MTV "Trade-off" (W3, Q3)**
>
> As discussed in Section 5, we openly acknowledge the trade-off regarding fine-grained nuances. However, this design is a necessary adaptation to the current domain constraints:
>
> * **Data Scarcity & Low SNR:** EEG–text datasets are tiny and extremely noisy compared to large-scale vision–language pretraining (e.g., CLIP's 400M pairs), making **posterior collapse** a real risk.
> * **Necessity of Summarization:** As shown in our ablation (Table 1), without MTV the model fails to capture even **coarse semantics**. Thus, MTV is essential under current conditions: it deliberately narrows the objective to stable semantic componets, rather than reconstructing inherently unstable verbatim details.
>
> ---
> ### **A6. Model Scalability & Architecture Choice (W4, Q4)**
>
> * **Plug-and-Play Design:** Our modular architecture is inherently plug-and-play. Swapping in larger language backbones is straightforward, and our internal experiments show the expected performance gains.
> * **Why Encoder–Decoder:** We favor Encoder–Decoder models (e.g., Flan-T5) because they naturally provide a structured, interpretable latent space for semantic alignment, as discussed in Related Work (Section 2).
> * **BCI Practicality:** In practical BCI settings, LLMs (mostly decoder-only models) are unnecessary. A compact model can reliably decode **core user intent** and elaborate it into fluent text, which is crucial for low-latency, edge-device deployment.

---

### Official Review · Reviewer_QM1m · 2025-11-01

**Soundness:** 2
**Presentation:** 3
**Contribution:** 2
**Rating:** 2
**Confidence:** 4

**Summary:**

This paper aims to address the posterior collapse in EEG-to-text decoding, i.e., mismatch between EEG signals and stimulus text. The authors proposed Generative Language Inspection Model (GLIM)—a framework to reframe the EEG-to-text decoding task as semantic summarization, rather than word-by-word reconstruction.
The authors address the posterior collapse issue with three technical components: (i) a contrastive-generative loss that aligns EEG embeddings to the frozen Flan-T5 latent space, (ii) multiple paraphrased text variants (MTV) per sentence to emphasise core semantics, and (iii) prompt-based domain adapters that cope with heterogeneous ZuCo recordings (different subjects, tasks, datasets). Extensive ablations show that the model produces fluent sentences, supports EEG-text retrieval, and achieves high zero-shot accuracy on sentiment/relation/corpus classification.

Despite the rich ablation study, the most prominent weakness is the complete absence of comparison with any other EEG-to-text algorithm except the 2022 EEG2Text baseline. Without results from recent methods such as DeWave, STG-decoder, or the contrastive-augmented models published at ACL/NeurIPS 2023-24, it is impossible to judge whether GLIM’s numbers represent a real advance or simply a different pipeline on the same data.

**Strengths:**

1. Novel problem framing: identifying posterior collapse as the root cause of hallucinations and recasting the task as semantic summarisation is original and interesting.
2. Rich ablation study: combining contrastive alignment, MTV data augmentation, and lightweight prompt adapters yields consistent gains in ablations.
3. Thorough self-diagnosis: the “noise-input” test and multi-view evaluation (generation, retrieval, zero-shot classification) demonstrate that the model actually listens to the EEG signal.

**Weaknesses:**

1．Missing SOTA baselines. Only EEG2Text is reported. Please include recent systems (e.g., DeWave, STG-based decoders, contrastive/MAE models from ACL/NeurIPS 2023–24) under the same split, or justify non-applicability and adapt where feasible.

2．Single-corpus evidence. All results are on ZuCo. Add cross-corpus tests (e.g., Natural Stories, UCLA Harry-Potter, Belt-2, ChineseEEG) to support generalization.

3．EEG encoder underuses neural structure. Temporal cross-attention downsampling overlooks spatial topology and frequency content, likely leaving discriminative information unused.

4．Retrieval is too easy. Top-1/Top-5 within 24-sentence batches inflates accuracy (even random vectors can exceed ~15% Top-1). No large-candidate retrieval (hundreds/thousands) or standard rank metrics (Median Rank, Recall@K, nDCG).

5．Noise-input evidence is insufficient. The model outperforming EEG2Text under noise doesn’t, by itself, prove hallucination robustness. Stronger controls (shuffled alignment, subject swap, graded noise) and semantic-aware metrics are needed.

6．Zero-shot comparisons feel inconclusive. Protocols do not clearly isolate semantic generalization attributable to GLIM vs. LM priors; make settings symmetric across models and report uncertainty.

7．MTV quality control is underspecified. The appendix lists prompts/rules but lacks quality validation for paraphrases; unclear whether “more MTV” is always better or where benefits saturate.

**Questions:**

suggestions
1．Add recent baselines. Report at least two modern EEG-to-text models (e.g., DeWave, an STG-style decoder, a contrastive/MAE variant such as Wang et al., ACL 2024) on the same train/val/test split.

2．Cross-corpus validation. Evaluate on Natural Stories, UCLA HP, Belt-2, ChineseEEG to test generalization.

3．Stronger EEG encoders. Explore graph CNNs over electrodes and frequency-band features; consider controlled partial LM fine-tuning.

4．Make MTV effects explicit. Vary the number/quality of paraphrases (0/1/3/5), show diminishing returns, and add BERTScore/BLEURT/MoverScore plus human ratings for semantic faithfulness/fluency.

5．Harder retrieval & proper ranking. Use candidate pools of hundreds/thousands; report Median Rank, Recall@{1,5,10,50}, and nDCG with CIs.

6．Robustness controls beyond one noise test. Add shuffled-pair, subject-swap, and graded-noise experiments; check for label/adapter leakage.

7．Fair zero-shot protocol. Standardize prompts, freezes, and adapters across models; report confidence intervals and paired significance tests vs. strongest baselines.

---

> ### Author Response · Authors · 2025-11-26
>
> We thank the reviewer for the detailed feedback. However, we have significant concerns regarding several **factually incorrect premises** in the review (e.g., chance-level math and nonexistent baseline/datasets). We address these points below.
>
> ---
> ### **A1. Baseline Comparisons (W1, Q1)**
>
> Requests to compare with "SOTA" models are addressed in the **General Response**. In short, the generic "STG-based decoders" and "contrastive/MAE models" are unspecified; while the DeWave and CET-MAE (Wang et al., ACL 2024) are not reproducible, rely on methodologically flawed generation (teacher forcing), and cannot serve as scientific baselines.
>
> ---
> ### **A2. Dataset Availability & Scope (W2, Q2)**
>
> The reviewer requests four datasets—only one actually exists.
> - **ChineseEEG**: real EEG dataset, but different language (Chinese vs. English) and reading paradigm (passive natural story reading vs. active reading with semantic-annoted sentences), **outside our scope**.
> - **"[Natural Stories](https://link.springer.com/article/10.1007/s10579-020-09503-7)"**: appears only as a text corpus rather than an EEG dataset.
> - **"[UCLA Harry-Potter](https://www.google.com/search?q=UCLA+Harry-Potter)"** can not be located in any public repository or publication;
> - **[Belt-2](https://openreview.net/forum?id=gp5dPMBzMH)**: a **model** evaluated on ZuCo, not a dataset.
>
>
> ### **A3. Model Design (W3, Q3)**
>
> - **Spatial Modeling**. The reviewer claims our encoder "overlooks spatial topology" and suggests us to explore GNNs for fully utilizing the "neural structure" information. We respectfully disagree with this idea. EEG electrode topology is highly inconsistent across subjects, equipment, and setups. Predefined graphs risk encoding incorrect structure. GLIM instead uses domain prompts and fully-connected layers for data-driven spatial adaptation.
>
> - **"Stronger" Encoder**: Suggestions such as frequency features or LM fine-tuning are appreciated. These directions become meaningful **after** establishing a reliable benchmark—which is precisely what GLIM provides.
>
> ### **A4. Retrieval Chance Level & Ranking Metrics (W4, Q5)**
>
> The reviewer claims random vectors can yield “~15% Top-1” among 24 candidates. This is mathematically false.
> - True chance-level is 1/24 = 4.17%, exactly as shown in Table 1 (row 3).
> - GLIM achieves **8.15%**, roughly **2x** chance. This is a **modest** improvement, and rank-based metrics would not add meaningful insight at the current performance level.
>
> ### **A5. MTV Quality & Analysis (W7, Q4)**
>
> - We provided 8 variants per sentence to ensure semantic coverage. Please refer to Appendix B.1, where we detail the rewriting rules and prompt templates.
> - The "diminishing returns" of variant count is a known phenomenon in augmentation; we selected 8 as a balance between diversity and computational cost. Further ablation on "number of paraphrases" (0 vs 8) is already provided in Table 1 (w/o MTV row), showing a clear benefit.
> - We agree that a deeper quantitative analysis of MTVs is valuable—MTVs have the potential to act as a **measurable model of semantic processing** (e.g., more human-like paraphrasing could correlate with stronger decoding performance). However, given current data limitations, computational budgets, and the performance ceiling, such fine-grained analysis is not yet feasible.
>
> ### **A6. Evaluation Sufficiency (W5, W6, Q4, Q6, Q7)**
>
> From  **Strengths 2 and 3** , the reviewer explicitly recognizes our "rich ablations" and "thorough self-diagnosis". Yet most subsequent critiques argue that our evaluation is **insufficient**—a direct logical contradiction.
>
> Even setting aside this inconsistency, the volume of extra demands is perplexing. An analogy may help:
>
> > You order a \\$5 cheeseburger. It arrives with fresh Angus beef, real cheese, and crisp lettuce. After praising its quality, you then complain that it doesn’t include prime rib, a full salad, and twenty slices of cheese. Meanwhile, the places you used to buy from only served two dry slices of bread with stale cheese for the same \\$5.
>
> This also mirrors the situation in our field:  **prior EEG-to-text studies provide far fewer evaluation than ours**.
>
> For anyone genuinely concerned about methodological reliability in generative brain decoding, we encourage a careful read of the paper and reproduction of our code.

---

### Official Review · Reviewer_g2vy · 2025-11-01

**Soundness:** 3
**Presentation:** 3
**Contribution:** 2
**Rating:** 4
**Confidence:** 3

**Summary:**

This paper presents GLIM, a generative model for EEG-to-text decoding that emphasizes semantic interpretability. The method introduces an intermediate “semantic subspace” to align EEG representations with textual embeddings via contrastive learning, followed by an autoregressive text generator that produces semantically faithful rather than word-by-word reconstructions of the original stimuli. Experiments on standard EEG-to-text datasets show moderate improvements in BLEU and embedding-based metrics, and the authors claim that GLIM captures interpretable linguistic features in its latent representations.

**Strengths:**

1.	The paper is exceptionally clear and well organized, making it easy to follow both the modeling approach and its neuroscientific motivation. It is a model example of strong writing and structure in the EEG decoding literature.
2.	The shift in objective—from literal text reconstruction to capturing the core semantic content of EEG—is conceptually meaningful and addresses an important limitation in previous EEG-to-text work. The attempt to quantify “semantic faithfulness” through embedding-based evaluation is well motivated.
3.	The authors support their claims with both decoding metrics and embedding analyses, and the correlation between latent representations and linguistic categories is an interesting exploratory result.

**Weaknesses:**

1.	The methodological novelty is limited. The proposed “semantic subspace” and training objectives largely reuse existing alignment and generation strategies, and the paper introduces no fundamentally new algorithmic component. Its contribution lies primarily in combining these techniques into a coherent and well-presented EEG-to-text framework.
2.	The distinction between literal decoding (“word-by-word reconstruction”) and semantic summarization is not fully explained. It is unclear how the model operationally achieves this shift—through multiple textual variants, specific loss weighting, or another mechanism. The two introduced losses (contrastive and generative) appear standard, and the results do not clearly demonstrate a qualitative difference in output type.
3.	Performance improvements over prior work are limited. While interpretability is claimed as the main advance, the supporting analyses remain descriptive and lack rigorous quantitative validation. More systematic testing—e.g., human judgments or controlled semantic perturbations—would strengthen the argument.

**Questions:**

1. The paper claims that GLIM shifts from literal reconstruction of the stimulus text to generating semantically faithful summaries. Could the authors clarify what concrete modeling choices or data settings enable this transition, and how this differs in practice from prior EEG-to-text approaches?
2. How is the “semantic subspace” enforced to capture interpretable linguistic dimensions, rather than merely acting as a dimensionality bottleneck? Have you explored direct supervision (e.g., via linguistic features) or ablations on this design?
3. Given the modest quantitative gains, what qualitative differences (e.g., in generated content or linguistic diversity) convince you that GLIM provides semantically richer decoding? Examples or human evaluation results would help make this case.

---

> ### Author Response · Authors · 2025-11-26
>
> We thank the reviewer for recognizing the clarity of our writing and the significance of our neuroscientific motivation. We address all the concerns below.
>
> ---
> ### **A1. Clarification on Methodological Novelty (w1)**
> The reviewer commented that our neural network components are "standard", and thus the "methodological novelty is limited". We respectfully argue that in applied deep learning studies, novelty lies not merely in inventing new mathematical operators, but in **identifying the root cause of domain-specific failures and designing the right architectural alignment to solve them**.
>
> GLIM is not a random combination of modules; on the contray, it offers a tailored whole-pipeline solution for semantically faithful EEG-to-text decoding. The novelty lies in this **neuro-linguistic alignment strategy** that successfully extracts semantic signals where previous studies fail.
>
> ---
> ### **A2. Operationalizing "Semantic Summarization" (w2, q1)**
> The reviewer asked how the shift from "literal reconstruction" to "semantic summarization" is operationally achieved. This is enforced through two mechanisms:
>
> - **Data-Driven Constraint (MTV):** By mapping a single EEG input to multiple paraphrased text variants (e.g., "The movie is great" / "It's a fantastic film"), we force the model to abandon one-to-one lexical memorization. Since the surface form varies, the model **must** learn the invariant "core semantics" to minimize the loss, effectively performing summarization rather than verbatim reconstruction.
> - **Latent Space Regularization:** The contrastive loss aligns the EEG representation with the **frozen, instruction-tuned latent space** of Flan-T5. Since this space is inherently structured by semantics (see Section 2), the EEG encoder is forced to output "meaning-rich" embeddings rather than just driving a token-by-token generator.
>
> ---
> ### **A3. Interpretability of the Semantic Subspace (q2)**
> Regarding how the subspace is "enforced" to be interpretable:
>
> - Since integrated language model (i.e., Flan-T5) is instruction-tuned on massive text corpora, its latent space is already highly organized and interpretable.
>
> - By aligning EEG to this **existing** topology (via parameter freeze and contrastive learning), the EEG representations automatically inherit the interpretability of the language model. This is verified by our consistently successful **Zero-shot Classification** on both representations and genrated texts (Table 1, 2), which would be impossible if the subspace were merely a "dimensionality bottleneck."
>
> ---
> ### **A4. Performance and "Modest" Gains (w3, q3)**
>
> The reviewer noted that improvements are "limited over prior work". We urge the reviewer to interpret these results in the context of the **"SOTA Illusion"** discussed in our **General Response**.
>
> Besides, we would like to emphasize GLIM's **significant semantic gains**. In terms of semantic decoding—which is our stated goal—the gains are substantial, not modest. We achieved 43%, 32%, 93% **zero-shot classification accuracy** across sentiment, relation and corpus (Table 1), far exceeding the chance-level baselines (33%, 11% and 50%). This confirms that while the open-ended generation is hard to boost due to the data limitation, the **semantic fidelity** has been significantly improved.

---

### Official Review · Reviewer_JPgs · 2025-11-02

**Soundness:** 3
**Presentation:** 3
**Contribution:** 3
**Rating:** 6
**Confidence:** 4

**Summary:**

This paper addresses hallucination in EEG-to-text decoding by reframing the task as semantic summarization rather than verbatim reconstruction. The authors propose GLIM (Generative Language Inspection Model), which learns interpretable EEG representations aligned with a frozen pretrained language model's latent space. The framework combines: (1) a contrastive-generative training objective combining language modeling and cross-modal contrastive learning, (2) multiple paraphrased text variants to promote semantic robustness, and (3) domain-prompt injection for heterogeneous data handling. Evaluated on the ZuCo dataset, GLIM demonstrates performance improvements in EEG-text retrieval, zero-shot semantic classification across sentiment/relation/corpus categories, and fluent sentence generation without teacher forcing.

**Strengths:**

1. The modular, plug-and-play architecture with minimal preprocessing enables scalability.
2. Original problem reframing tied to a principled failure mode (posterior collapse) with a concrete mitigation (Sec. 2–3).
3. The three-pronged evaluation (generation, retrieval, classification) provides much stronger validation than previous work relying solely on BLEU/ROUGE scores.

**Weaknesses:**

1. Main results appear single-run; no confidence intervals/seed variance for Table 1. Authors should report mean+CI over ≥3 seeds for all metrics, including controls.
2. The paper has limited technical novelty. Core components (Q-former-style alignment, contrastive learning, domain prompts) are borrowed from existing work. The contribution is primarily in their combination for this specific task.
3. Semantic evaluation (zero-shot classification) relies on pretrained LM priors; unclear whether improvements reflect EEG alignment or LM bias. Maybe try label-agnostic MTV controls, shuffled/alternative label embeddings (or weaker/random LM encoders) to support the results.
4. Insufficient baseline comparison. EEG2Text results are from Jo et al. (2024) using a different data split with potential train-test overlap (noted by †). This severely limits the validity of performance comparisons. The authors should reproduce EEG2Text on their split, and include additional brain decoding baselines for a fairer evaluation
5. The notion of "core semantics" is not rigorously defined. The current evaluation captures only coarse categorical semantics and omits finer dimensions such as paraphrase quality, factual accuracy, and semantic similarity. The authors should include or discuss additional semantic evaluation metrics (e.g., BERTScore)

**Questions:**

1. Could you report per-subject or per-task variance to confirm statistical reliability?
2. How do you ensure that MTV paraphrases do not leak information about sentiment/relation labels via LLM priors?
3. Can you reproduce EEG2Text on your exact data split to enable fair comparison? The current comparison is inconclusive.
4. Have you validated that the paraphrased variants actually preserve semantic content through human annotation? How do you ensure variants don't introduce systematic biases?
5. Why these specific semantic categories (sentiment, relation, corpus)? Have you tested other categories like named entities?

---

> ### Author Response · Authors · 2025-11-25
>
> We appreciate the reviewer for the positive feedback and insightful questions. We address all the concerns below.
>
> ---
> ### **A1. Result Reliability (w1, q1)**
> Please refer to our **General Response**, where we discuss **internal consistency** and our **process transparency** as the primary evidence for reliability in this benchmark study.
>
> ---
> ### **A2. Clarification on MTV Label Leakage (w3, q2)**
> We clarify that there is **no label leakage** in our framework.
> - **Input Isolation:** As detailed in Section 3 and Figure 2, the model inputs strictly consist of EEG signals and domain prompts. The domain prompts serve only for domain adaptation and **do not contain any semantic labels**, as verified in Section 4.4 and Table 3.
> - **Proof via Noise Input Test:** The strongest proof against leakage is our **noise input test**. As shown in Table 1 (Row 5), when EEG is replaced by noise (while keeping any other variable unchanged), the performance drops significantly to chance level. If the MTV had leaked labels into the pipeline in some way, the model would have maintained high performance even with noise input.
>
> ---
> ### **A3. Rationale for Baseline Selection and Comparison (w4, q3)**
> We did not perform a full reproduction of the EEG2Text model for 3 reasons:
> - **Metric Incompatibility:** GLIM prioritizes interpretable, zero-shot evaluation in a **frozen LM-aligned representation space**. EEG2Text is a fully fine-tuned end-to-end model; it lacks a comparable, semantically structured embedding space for our primary representation metrics (Table 1).
> - **Poor Generation Quality:** As reported in [Jo et al., 2024](https://arxiv.org/abs/2405.06459), the EEG2Text model tends to generate repetitive patterns (e.g., starting every sentence with "He was...") that fail to distinguish even basic corpus topics. Reproducing it on our stricter split without extensive hyperparameter tuning would likely yield trivial results that offer little comparative value.
> - **Methodological Flaw:** As analyzed in Section 1, the failure of EEG2Text stems from **Posterior Collapse**—the mismatch between the powerful autoregressive loss and the low information capacity of EEG. Our ablation (Table 1, Row 6) confirms that relying solely on autoregressive loss (like EEG2Text) leads the model to ignore EEG inputs entirely. Therefore, while we acknowledge the modified EEG2Text as the only "honest" (non-teacher-forcing) baseline available, its inherent methodological flaws make it unsuitable for a comprehensive comparison.
>
> ---
> ### **A4. Quality and Bias Control in MTV (q4)**
> - **Quality Assurance:** Current LLMs have demonstrated high reliability in instruction following. We performed extensive prompt optimization and preliminary testing before generating the final MTV set (see Appendix B.1). All generated variants are included in the supplementary material for transparency.
> - **Robust Semantic Guide:** We emphasize that MTVs are designed to function as a **collective semantic guide**. As discussed in Appendix B.2, even if individual variants contain minor noise, the model learns to extract the common core semantics shared across the set, effectively filtering out outliers during training.
> - **Bias Mitigation:** To prevent systematic bias, we explicitly designed **three orthogonal rewriting strategies** and utilized two different models (LLM vs. Integrated LM), resulting in 8 diverse variants per sample. This structural diversity ensures that the model aligns with the abstract semantic content rather than overfitting to any single model's bias.
>
> ---
> ### **A5. "Core Semantics" and Evaluation Granularity (w5, q5)**
> The "core semantics" in our context refers to coarse-grained semantic intent (e.g., sentiment, relation, topic) rather than fine-grained details. This definition is grounded in both result observations and data validity:
>
> * **Performance Reality:** GLIM achieves moderate accuracy on these coarse categories. Decoding finer-grained semantics currently exceeds the sysmetic ceiling.
> * **Label Validity & Active Processing:** We argue that the validity of semantic labels depends on the reading paradigm. In "Active Reading" (TSR in ZuCo), subjects must answer comprehension questions, ensuring the brain actually processes the semantic content. In "Passive Reading" (NR), subjects may merely skim the text without deep semantic encoding. Our results support this: the TSR group consistently yields better generation diversity and semantic accuracy, suggesting that valid "core semantics" are only reliably encoded when subjects are actively engaged.
> * **BCI Utility:** From an application perspective, we prioritize the reliability of representation decoding over verbatim reconstruction. If a BCI can accurately decode the core semantics (user intent), a backend LLM can easily utilize context to flesh out a fluent, complete sentence. Thus, our evaluation focuses on whether the intent is captured, rather than sentence-level similarity.

---

### Author Response · Authors · 2025-11-25
**General Response: Establishing a Transparent and Reliable Open Benchmark**

We thank the reviewers for their constructive feedback. To address concerns regarding **baseline comparisons** (Reviewer QM1m, pGpW) and **result reliability** (Reviewer JPgs), we clarify that the core contribution of this work is to establish a **transparent, reproducible, and methodologically rigorous benchmark/protocol** for EEG-to-text decoding—a field currently facing challenges with unreproducible results and "teacher-forcing" illusions.

---
### **1. The Challenge: Unreproducible and Methodologically Flawed "SOTAs"**

Reviewer QM1m and pGpW suggested comparisons with recent "SOTA" models such as [DeWave](https://arxiv.org/abs/2309.14030) and CET-MAE ([Wang et al., 2024](https://aclanthology.org/2024.acl-long.393/)). We respectfully decline this comparison. By tracing the chronological development of this field, we clarify why these models do not constitute valid baselines.

The field's early study, **EEG2Text** ([Wang et al., 2021](https://arxiv.org/abs/2112.02690)), established a codebase that became the blueprint for many subsequent works. However, as admitted later by the authors (see their [GitHub repo](https://github.com/MikeWangWZHL/EEG-To-Text)), this model contained a critical flaw: it relied on **teacher-forcing** (**TF**) during inference, where predictions were conditioned on ground-truth text labels. This issue was exposed by a recent study **"Are EEG-to-text models working?"** ([Jo et al., 2024](https://arxiv.org/abs/2405.06459)). In this study, they demonstrated that when tested without TF, the original EEG2Text model achieved similar BLEU/ROUGE scores given **pure random noise** inputs as it did with EEG inputs. This proved that the high scores reported in the literature were largely hallucinations driven by the language model prior, not decoding from brain signals.

Despite this revelation, many subsequent works (referenced in [NeuSpeech/EEG-To-Text](https://github.com/NeuSpeech/EEG-To-Text)) had already adopted the flawed EEG2Text codebase with TF settings, including the aforementioned DeWave and CET-MAE. While recent work like CET-MAE discussed the limitations of teacher-forcing, **valid "non-TF" benchmarks remain absent from the literature**. It is particularly telling that the first author of **DeWave** is also a co-author of the critical study (Jo et al., 2024). Yet, even in that rigorous critique, no reproducible "free-generation" results for **DeWave** were reported to refute the hallucination hypothesis. This strongly suggests that a valid SOTA baseline **simply does not exist** prior to our work.

That is to say, to the best of our known, although it yields low scores (reflecting the true difficulty of the task) and has unintentionally inherited data leakage (see our Appendix A.4), Jo et al's modified EEG2Text is the only available model for our comparsion. GLIM is designed to rise from this honest foundation, rather than competing with inflated scores from unverifiable or methodologically flawed models.

---
### **2. The Solution: Ensuring Reliability via Process Transparency**

For the reliability concern (Reviewer JPgs), we argue that reliability is best ensured through **codebase transparency** and **strict process determinism,** rather than statistical averaging of opaque runs. First, we have already released our complete codebase, including the data splitting notebooks, all generated samples and the model checkpoint. Second, we carefully fixed all random seeds (e.g., for data splitting and training batch shuffling) and trained all models with a fixed training duration (200 epochs) to maximize the training reproducibility.

By reporting the result of this fixed, transparent pipeline without manual checkpoint selection (i.e., "cherry-picking"), we provide a reproducible lower bound of the method's capability—which allows full community verification and benchmarking.

---
### **3. The Proof: Internal Consistency in Complementary Evaluation**

The reliability of our results is strongly evidenced by the **internal consistency** observed across orthogonal dimensions, which refutes the possibility of random fluctuation.

- **Structural Consistency across Metrics:** As demonstrated in Section 4, our proposed GLIM improves performance consistently across **generation**,  **retrieval**, and  **zero-shot classification**. Conversely, **ablating components** or replacing EEG input with random noise (i.e., the **noise input tests**) leads to a systematic drop across all these metrics. Such structural synchronization across diverse tasks is a strong indicator of model validity.

- **Inter-Subject & Domain Consistency:** As shown in  **Figure 5, Appendix D**, GLIM exhibits stable performance distributions across different subjects and domains. Specifically, the pattern of intra-domain clustering (across subject dots) and inter-domain decoupling (between dataset versions, reading paradigms and corpora) confirms the model captures robust, subject-invariant semantics from brain signals.

---

### Author Response · Authors · 2025-12-04
**Beyond Benchmarks: The Vision of Practical Language BCIs**

**To the Reviewers, ACs, PCs and the Broader Research Community**:

Given the recent procedural disruptions in the review process, we recognize the limitations of detailed technical rebuttals at this stage. Instead, we write this note to clarify the **long-term vision** behind GLIM and to advocate for a paradigm shift in how we approach non-invasive language Brain-Computer Interfaces (BCIs).

GLIM is not merely an algorithmic improvement; it is a **strategic step** toward a future where non-invasive neural interfaces become part of everyday interaction. We believe the path to **Practical Language BCIs** lies in the symbiosis of **Neural Intent Decoding** and **Contextual AI**.

---
### **1. The Paradigm Shift: From "Text Reconstruction" to "Intent + Context"**

The field of EEG-to-text has long been trapped in the pursuit of verbatim reconstruction, a goal that often leads to "hallucinations" given the low SNR of non-invasive signals. We argue that the future lies in a different architecture, driven by the convergence of hardware and software:

- **The Neural Component (The "Anchor")**: The brain's role is not to dictate every token, but to provide the **Core Semantic Intent**. This is why GLIM reframes the task as "Semantic Summarization" and insists on rigorous, non-teacher-forcing evaluation to ensure the decoded signal is genuine. GLIM acts as the trustworthy "Intent Anchor."

- **The AI Component (The "Sail")**: In the age of Agentic AI, a system equipped with a **User Context Graph** (environmental awareness, dialogue history) can reconstruct fluent, precise language from a coarse neural intent. Just as modern voice interfaces (e.g., [Wispr Flow](https://wisprflow.ai/)) transform humble whispers into coherent text using context, **future BCIs will transform "neural whispers" into fluent communication**.

---
### **2. Why Rigor Matters for this Future**

This vision cannot be built on sand. The current prevalence of unreproducible baselines and "teacher-forcing illusions" in the community threatens to derail progress.

- **Integrity as a Prerequisite**: We adhered to a strict, anti-cherry-picking protocol and released verifiable code because **scaling laws** can only be established on honest data.

- **The Roadmap**: GLIM validates this **"Intent Decoding"** capability on research-grade data (ZuCo). This verification is the necessary prerequisite before we can confidently **scale to massive, noisy data from emerging wearable EEG devices** (e.g., [neural headbands](https://www.sciencedirect.com/science/article/abs/pii/S0010482524015488), [earbuds](https://patents.google.com/patent/US20240008800A1/en), [smart glasses](https://ieeexplore.ieee.org/document/10780518)).

---
### **3. A Call for Collaboration**

We are actively exploring the intersection of **Brain-to-Text Scaling Laws** and **Context-Aware Agents**. If you are an individual or organization (such as teams working on wearable neural interfaces or context-aware input solutions) who shares this vision of building the next generation of human-computer interaction, we invite you to connect with us.

We hope GLIM serves not just as a benchmark, but as a foundation for this exciting future.

---

### Meta-Review · Area_Chair_K87W · 2025-12-26

**Summary:**

The paper focuses on hallucination in EEG-to-text decoding by redefining the task as semantic summarization instead of verbatim reconstruction. It introduces GLIM (Generative Language Inspection Model) that learns interpretable EEG representations aligned with a frozen pretrained language model's latent space. Experiments show that GLIM achieves performance improvements in EEG-text retrieval, zero-shot semantic classification.
While well presented, its experiments and technical novelty are considered insufficient to justify acceptance at this time.
Considering the reviewers’ concerns, we regret that the paper cannot be recommended for acceptance at this time. The authors are encouraged to consider the reviewers’ comments when revising the paper for submission elsewhere.

**Reviewer Concerns:**

Key concerns include (1) Experiments (Main results appear single-run and the reviewer suggests to report mean+CI over ≥3 seeds for all metrics, including controls), (2) Limited technical novelty (Core components (Q-former-style alignment, contrastive learning, domain prompts) are borrowed from existing work.).

**Reviewer Scores:**

Scores ranged from Reject, marginally below to marginally above the acceptance threshold, with multiple reviewers explicitly stating they would not object to rejection.

---

### Decision · Program_Chairs · 2026-01-26

Reject